# Longitudinal long-read microbiome profiling in a canine model reveals how age, diet, and birth mode shape gut community dynamics

Md Asaduzzaman,[1,2] Péter Oláh,[2,3] Natheer Jameel Yaseen,[1,2] Ahmed Taifi,[1,2] Tamás Járay,[1,2] Gábor Gulyás,[1,2] Zsolt Boldogkői,[1,2] Dóra Tombácz[1,2]

**ABSTRACT**  The gut microbiome undergoes dynamic age-related changes shaped by diet and maternal factors. Here, we present a species-level, long-read 16S rRNA survey of the developing gut microbiome in a translational canine model, profiling 89 purebred Hungarian Pumis across early-life and reproductive stages. We collected 456 fecal samples longitudinally: 60 puppies followed from birth to 81 weeks, their mothers sampled during pregnancy and lactation, and adult controls from six kennels. We recorded detailed dietary metadata and reproductive status throughout the study. Age was the strongest determinant of alpha diversity, with a rapid increase during weaning and stabilization by 6 months of age. Beta diversity analyses revealed structured compositional transitions from early developmental phases to adulthood, including a shift toward more uniform, adult-like communities. Within-kennel variation was modest, consistent with shared environmental exposures. Mixed-effects models showed robust associations between specific taxa and age, diet, and kennel, while SparCC-inferred co-occurrence networks indicated increasing ecological complexity with age. We also demonstrated that the delivery mode—vaginal versus cesarean—impacted early-life microbiome composition: *Lactobacillus* spp. were significantly more abundant in cesarean-born puppies than in vaginally delivered littermates during the 8–10-week window. We also observed reproducible maternal microbiome shifts during pregnancy and lactation, with potential implications for vertical microbial transfer. Taken together, our results show that domestic dogs follow a reproducible, age-structured trajectory of microbial maturation that parallels human development, including delivery-mode effects and diet-responsive taxa.

**IMPORTANCE**  Microbiome research is among the fastest-moving areas in biomedicine driven by major global efforts to understand how microbial communities shape human health and disease. Dogs provide an ideal translational model because their gut microbiota more closely resembles that of humans than that of other studied animals; moreover, breeds show high within-breed genetic homogeneity; diets can be tightly regulated; and longitudinal sampling across the lifespan is feasible. Mapping shifts driven by diet and maternal factors—from early-life events through later life, including senior stages—is essential to leverage microbial plasticity for prevention, with implications for inflammation, metabolic disease, and neurodegeneration. Here, we advance this goal by providing a longitudinal, high-resolution data set and demonstrating that full-length 16S rRNA sequencing is a powerful tool for resolving fine-scale patterns of gut colonization and maturation.

Address correspondence to Dóra Tombácz, tombacz.dora@med.u-szeged.hu.

The authors declare no conflict of interest.

See the funding table on p. 15.

10.1128/msystems.01279-25 **1**

**KEYWORDS** canine gut microbiome, long-read 16S rRNA sequencing, longitudinal study, microbiome maturation, birth mode, maternal microbiome, weaning and diet, alpha and beta diversity, species-level resolution, co-occurrence networks

The gut microbiota comprising bacteria, archaea, viruses, fungi, and protozoa plays a critical role in host immunity and metabolism via the production of metabolites that influence both intestinal and systemic health (1, 2). The collective genomes of these microorganisms constitute the gut metagenome. How these communities assemble and stabilize in early life remains a major focus of microbiome research. Dogs (*Canis lupus familiaris*) are valuable comparative and translational models because they share human environments and exhibit human-like dietary patterns, physiology, and disease risk profiles (3). Notably, compared with pigs or mice, canine gut microbiomes are more similar to those of humans in species-level composition and diet-responsive dynamics (4), supporting their relevance for cross-species investigations of host-microbiome interactions.

Canine gut microbiome composition varies with age, diet, reproductive status, and genetics (5). In older age, diversity decreases; taxon abundances shift (6–9); and dysbiosis emerges (10); this trajectory coincides with immunosenescence and chronic low-grade inflammation ("inflammaging") (11–13). Although the human infant microbiome is well characterized, comparable longitudinal canine data remain limited (14, 15). Previously, the mammalian fetal gut was considered sterile (16), but accumulating molecular evidence has challenged the "sterile womb" hypothesis, with reports of microbial DNA in the placenta, amniotic fluid, and meconium (17, 18). However, the existence and extent of a true prenatal microbiome remain controversial, and several authors have argued that many of these low-biomass signals may reflect contamination rather than stable fetal colonization (16, 17, 19).

In dogs, microbial DNA has been detected in meconium and amniotic fluid following both vaginal and cesarean deliveries (20, 21). Vertical transmission occurs via birth, nursing, and close maternal contact (22). In humans, delivery mode has a major impact on early colonization dynamics and microbial diversity (23–26), but corresponding canine studies remain scarce (20, 21).

Diet is another key modulator of gut microbiota composition. Although classified as facultative carnivores, domestic dogs can adapt to a wide range of dietary patterns. Most commercial products are high in carbohydrates and are extruded into dry kibble, while raw and high-protein formulations offer alternative macronutrient profiles (27). Dietary interventions modulate microbiota composition and metabolic outputs, with downstream effects on inflammation and health status (4, 28). Several studies have shown that the canine microbiome responses to diet resemble those observed in humans, particularly with respect to macronutrient-driven shifts (4, 29, 30). For example, both protein and fiber content influence gut taxonomic and functional profiles (29). Despite these insights, key knowledge gaps remain in our understanding of microbial development, vertical transmission dynamics, and diet-microbiome interactions during early life (15, 31, 32). In particular, the temporal structure of microbiome maturation and the impact of birth mode or maternal status in dogs are not well characterized. Moreover, although diet is a well-known modulator of adult gut communities, its role across different stages of postnatal development is less understood (30, 33, 34).

Breed-specific microbiota variation has been reported across mammalian species (35–38). In dogs, *Firmicutes* and *Bacteroidetes* consistently dominate the gut microbiota across breeds and physiological contexts (35, 39–41). However, few studies have investigated how breed- or kennel-specific factors interact with developmental and dietary influences in shaping microbial succession. Longitudinal sampling is essential to capture temporal microbial dynamics and disentangle age-related effects from environmental variation (42). Full-length 16S rRNA gene sequencing enabled by long-read platforms, such as Oxford Nanopore Technologies (ONT) and Pacific Bioscien-

ces, provides species-level resolution and enables high-resolution taxonomic mapping across diverse hosts (43–47).

In this study, we present a longitudinal canine microbiome data set with species-level resolution, capturing the developmental trajectory of gut microbial communities from birth to 81 weeks of age in purebred Hungarian Pumi dogs. We also examine the effects of delivery mode, maternal status, and diet on community structure and explore community co-occurrence patterns across early life. This work aims to advance our understanding of postnatal microbiome assembly in a translational model species.

# RESULTS

## Study design and analytical pipeline

In this study, we profiled 456 fecal samples from 89 purebred Hungarian Pumi dogs—55 puppies followed from birth to 81 weeks, nine dams across pregnancy/lactation, and 20 time-matched adult controls—with a small cesarean-delivered subgroup (five puppies from one litter) analyzed separately. DNA was extracted (ZymoBIOMICS 96 MagBead); full-length 16S rRNA amplicons (V1–V9) were sequenced on MinION (R9) and basecalled with Dorado. Taxa were profiled with EMU and diversity computed in phyloseq from unfiltered counts. Downstream R analyses applied prevalence/abundance filters (>5%; ≥5 samples), mixed-effects models (lme4) for age, sex, kennel, kennel:litter, and diet (age modeled categorically and with splines: diets categorized based on FEDIAF guidelines), and comparative tests (Wilcoxon, $t$-test, Mann-Whitney; α = 0.05). Community structure was summarized by hierarchical clustering and heatmaps (ComplexHeatmap), taxonomic trees in iTOL, and SparCC co-occurrence networks (CONs; SpiecEasi) visualized in igraph/ Cytoscape (Fig. 1).

## Sequencing output

The MinION sequencing runs yielded a total of 121.30 million pass (Q ≥ 10) reads (mean ± standard deviation [SD]: 266,017.09 ± 127,742.50 per sample) with an average read length of 1,521.81 ± 50.07 bp and a mean Q-score of 20.40 ± 1.05. This corresponded to approximately 185.01 GB of raw data.

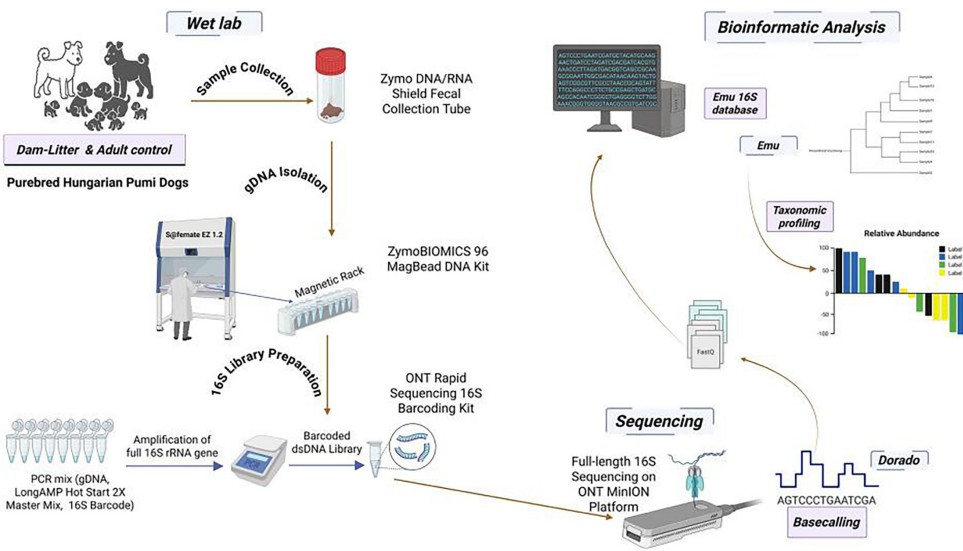

**FIG 1** Graphical abstract of the study workflow. Fecal samples from purebred Hungarian Pumi dogs were processed for full-length 16S rRNA gene sequencing on the Oxford Nanopore MinION platform, followed by Dorado basecalling and taxonomic profiling with the Emu 16S database.

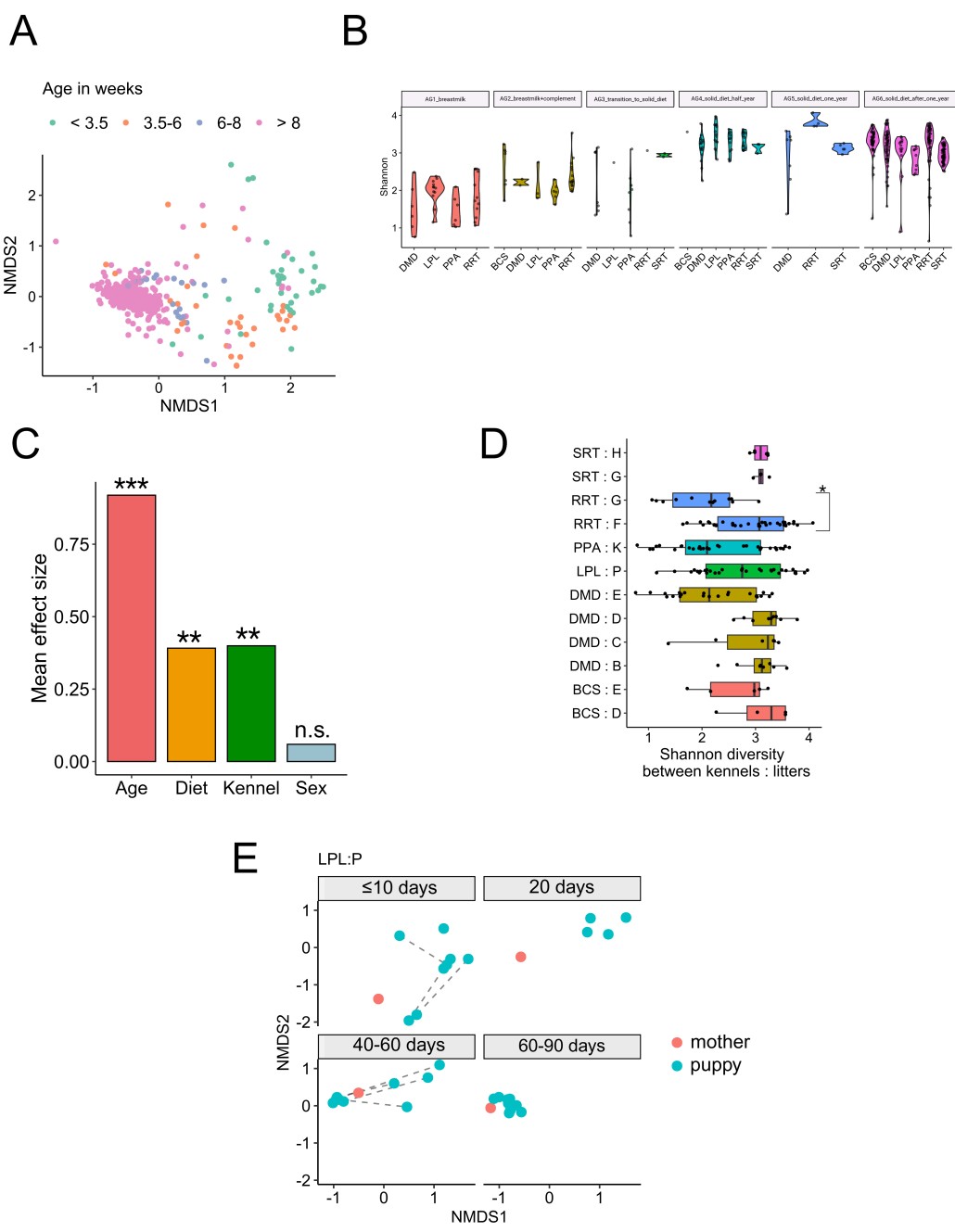

**FIG 2** Longitudinal diversity patterns and within-litter variation in canine gut microbiome. (A) NMDS plot revealing a developmental gradient in the gut microbiota with increasing homogeneity in adulthood. (B) Shannon diversity index across age groups showing microbial expansion during early development. Age groups: 1 (<3.5 weeks), 2 (3.5–6 weeks), 3 (6–8 weeks), 4 (10–26 weeks), 5 (26–52 weeks), and 6 (>52 weeks). (C) Relative effect sizes of cohort variables on alpha diversity (Shannon index). Analysis was based on a cumulative link model adjusted for covariates and subject ID (***$P < 0.001$; **$P < 0.01$; n.s.: non significant). (D) Within-kennel litter comparisons of Shannon diversity showing mostly modest differences with one significant exception (Wilcoxon test, *$P < 0.05$). (E) Beta-diversity distances within a single litter. Samples from the mother and pups are shown in red and blue, respectively. Dotted lines connect longitudinal samples from the same subject taken within the same age period.

## Microbial diversity increases during puppyhood but stabilizes in adulthood

Ordination of the sampled cohort by non-metric multidimensional scaling (NMDS) reveals a clear age-associated gradient (Fig. 2A): neonatal and early-life samples are

widely dispersed, indicating high inter-individual variability, whereas profiles from later time points cluster more tightly, consistent with progressive convergence toward an adult-like community. In line with this, alpha diversity—measured by the Shannon index—increases markedly during the first weeks of life, with a pronounced rise around the weaning window (3.5–6 weeks), followed by a gradual tapering and an apparent plateau by approximately 6 months of age (Fig. 2B). This trajectory is evident across kennels and age strata (age groups 1–6 as defined in Fig. 2B), indicating that the maturation pattern is robust to kennel-level differences in husbandry and environment.

## Effects of environment and host sex on community composition are modest

To quantify the relative contribution of host and environmental variables, we estimated effect sizes for alpha diversity using a cumulative link model adjusted for covariates and subject ID. Age emerged as the dominant predictor ($P < 0.001$), with kennel exerting a smaller but statistically significant effect and sex showing no detectable contribution (Fig. 2C). Comparisons between litters within the same kennel revealed generally modest differences in overall Shannon diversity, with only a single litter pair reaching statistical significance (Wilcoxon test, $P < 0.05$; Fig. 2D). Finally, within a representative litter, beta diversity distances relative to the dam show early divergence at ≤10–20 days, followed by increasing similarity by 40–60 days that remains stable at later intervals (Fig. 2E). Together, these patterns indicate that age-linked maturation—and the diet transitions that accompany it—are the primary drivers of community structure, while environmental variation at the kennel or litter level exerts comparatively smaller effects.

## Developmental stage shapes taxonomic composition

Analysis of the most abundant genera across litters and kennels identified clear compositional shifts over time (Fig. 3A). During the weaning transition (3.5–8 weeks), early-life taxa—*Escherichia*, *Megamonas*, *Lactobacillus*, and *Ruminococcus*—were dominant, whereas the post-weaning period was characterized by the higher relative abundance of *Blautia*, *Peptacetobacter*, and *Fusobacterium*. At early weaning (6–8 weeks), *Lactobacillus* exhibited kennel-specific presence and diet-linked abundance patterns (Fig. S1). Notably, *Blautia* was consistently abundant across all life stages and showed a steady rise over time (Fig. 3A), suggesting its establishment as a core post-weaning genus.

To identify the dominant members of the gut microbiome in the cohort, we examined the most prevalent and abundant species. As shown in Fig. 3B, *Blautia producta*, *Blautia hansenii*, *Faecalimonas umbilicata*, and *Megamonas rupellensis* were among the most prevalent species, while *Blautia glucerasea* and *Peptacetobacter hiranonis* exhibited the highest relative abundance. A taxonomic tree highlighting the top 350 taxa revealed distinct associations with age and kennel of origin (Fig. 3C). Species-level resolution enabled by full-length 16S sequencing captured fine-scale taxonomic patterns, including taxa showing consistent shifts across multiple litters and kennels. At the genus level, hierarchical clustering of the top 40 taxa showed that age and diet were more influential than sex in shaping microbial community structure (Fig. S2). To further explore host and environmental influences on taxonomic profiles, we performed mixed-effects modeling across four key factors: age, kennel, diet, and sex. Each model revealed distinct taxa significantly associated with each factor (Fig. S3 through S6).

## Species-level core microbiome across age

Using a prevalence-based definition, we identified a species-level core within each age group and present three complementary analyses. First, focusing on the strict 90%-prevalence criterion, a set of 13 species was present in all age groups; their mean relative abundances varied systematically with age (Fig. S7A). This ubiquitously detected set was dominated by short-chain fatty acid (SCFA)-associated *Clostridia* (e.g., *Blautia* spp., *Faecalimonas umbilicata*) (48) and the bile-acid-transformer *Peptacetobacter hiranonis*, which has been associated with maintenance of intestinal barrier integrity, colonization resistance against pathogens, such as *Clostridioides difficile*, and broader host metabolic

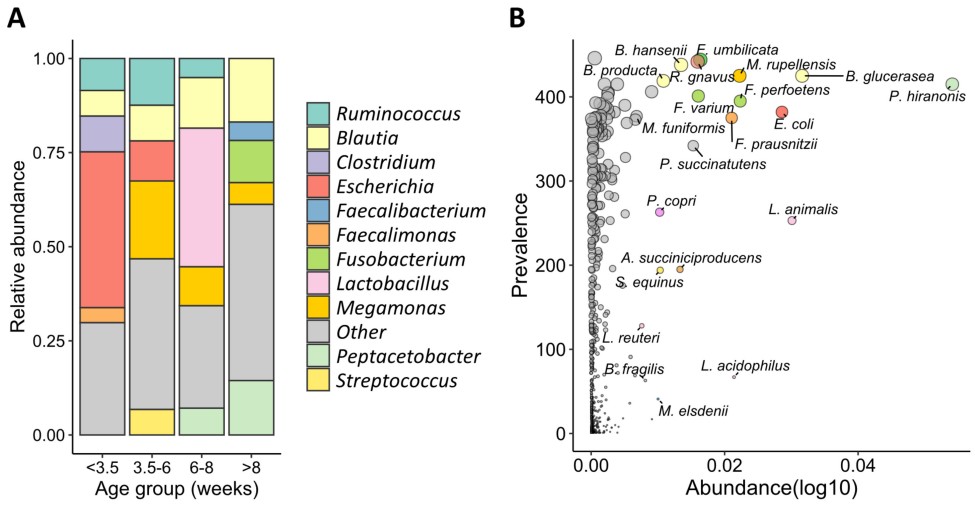

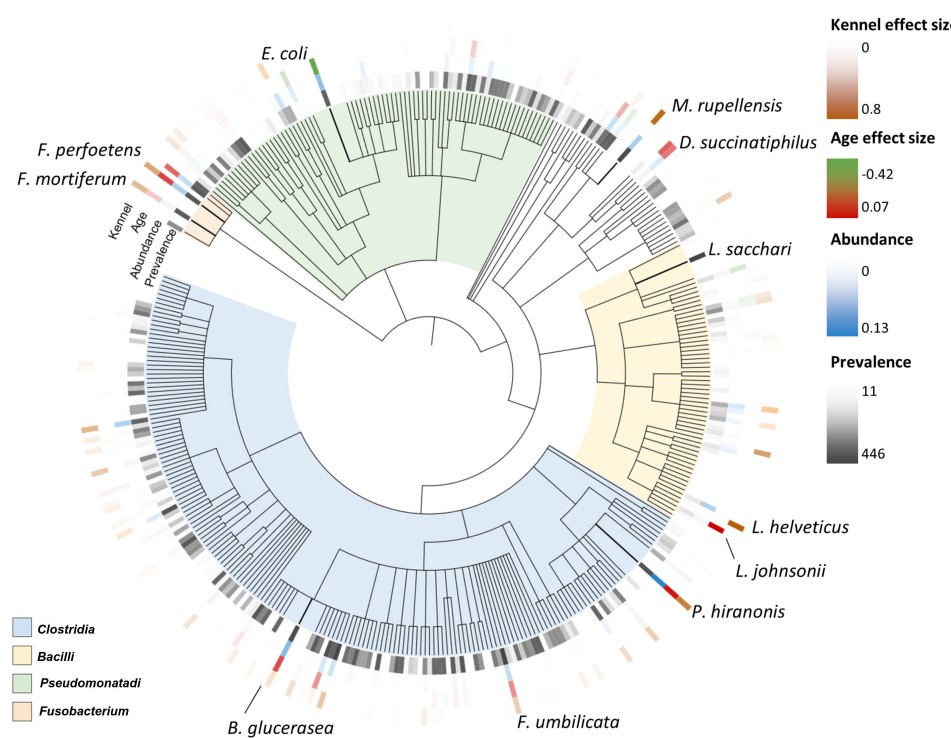

**FIG 3** Key taxa of the canine gut microbiome: abundance-prevalence patterns and age/kennel associations. (A) Bar plot of top genus relative abundances across kennels and litters divided by animal's life phase. (B) Scatter plot of the most abundant (*x*-axis) and most prevalent (*y*-axis) species across all samples in the cohort. Dot size is proportional to prevalence; dot color corresponds to genus for the top 25 data points and is harmonized with panel A. (C) Phylogenetic tree of the top 350 taxa identified across all kennels by prevalence and abundance. Mean effect sizes obtained via mixed-effect models of association with age and kennel are shown with contrasting color scales. Species with the highest values per category are highlighted.

homeostasis (49). Some taxa (e.g., *Megamonas rupellensis*) showed prominence in both early and late lives, aligning with shifts in diet and overall community complexity.

Second, pairwise Jaccard similarities between 90%-prevalence core sets (Fig. S7B) were highest between adjacent late-life stages (e.g., 3–7 years vs. 7–10 years), indicating a stable, adult-like core with gradual turnover, whereas overlaps between pre-weaning and adult groups were minimal, highlighting weaning as a major ecological breakpoint.

Third, to visualize broader intersections, we applied a more inclusive 80%-prevalence threshold in an UpSet analysis (Fig. S7C). The largest intersection corresponded to the same 13 universally present species, while multiple intermediate intersections spanned two to five adjacent age groups, indicating progressive rather than abrupt replacement during maturation.

### Functional context

The persistent core is enriched for SCFA producers and bile acid modulators, which contribute to mucosal integrity and immune-metabolic balance across life stages; early-life deviations and late-life shifts likely reflect changing dietary substrates, host physiology, and age-related reductions in community redundancy.

### Developmental rewiring of the microbial network

SparCC-based CONs revealed striking differences in microbial interactions across age groups (Fig. 4). In pups younger than 3.5 weeks, the network was sparsely connected with few co-occurring taxa. At 6–8 weeks, network density and modularity increased, indicating more structured microbial communities. By 1–2 years of age, the adult microbiome exhibited robust and highly connected networks, with strong positive associations (red edges) among core taxa, such as *Bacteroides*, *Clostridium*, and *Sutterella*. These transitions are consistent with compositional and functional maturation of the gut microbiota.

### Birth mode influences microbiome composition during early postnatal development

A focused comparison of two litters from the same dam and kennel (Le Petit Lapin)—one delivered by cesarean section and one by vaginal delivery—matched for age (8–10 weeks) and diet revealed differences in beta diversity (NMDS ordination) between the groups (Fig. 5A). However, Shannon alpha diversity did not differ significantly (Fig. 5B). Notably, C-section pups showed higher relative abundances of *Lactobacillus* (*P*.adj = 0.008) and *Prevotella* (*P*.adj = 0.038) and a lower abundance of *Romboutsia* (*P*.adj = 0.045) than vaginally delivered pups (Fig. 5C and D). These findings suggest that delivery mode can leave a lasting imprint on early gut colonization under otherwise controlled conditions. Contrary to many human studies—where C-section is associated with reduced *Bacteroides*/*Prevotella* and lower *Lactobacillus* early in life—our C-section pups showed higher *Lactobacillus* and *Prevotella* at 8–10 weeks.

### Maternal gut microbiome shifts with reproductive stage

The gut microbiota of dams displayed transient reductions in alpha diversity during late pregnancy and early lactation, followed by partial restoration post-weaning (Fig. 6A). Despite temporal proximity, microbiome profiles on the prepartum and postpartum days exhibited distinct compositional signatures (Fig. 6B). The transition from prepartum to postpartum was marked by clear taxonomic turnover, with the disappearance of *Blautia* and *Fusobacterium* and the emergence of *Clostridium* and *Streptococcus* (Fig. 6B). These findings highlight the dynamic microbiome shifts associated with reproductive status and their potential role in seeding the neonatal microbiota. On the contrary, while the core microbiome composition of mother dogs significantly shifts during reproductive stages, these shifts are not associated with specific kennels or diets and are not consistent across individuals, as evidenced by unsupervised clustering (Fig. 6C).

### DISCUSSION

This longitudinal, species-resolved survey of purebred Hungarian Pumi dogs shows that gut microbiome maturation is a strongly age-linked process that proceeds through an early, variable phase toward a more homogeneous adult community. Alpha diversity

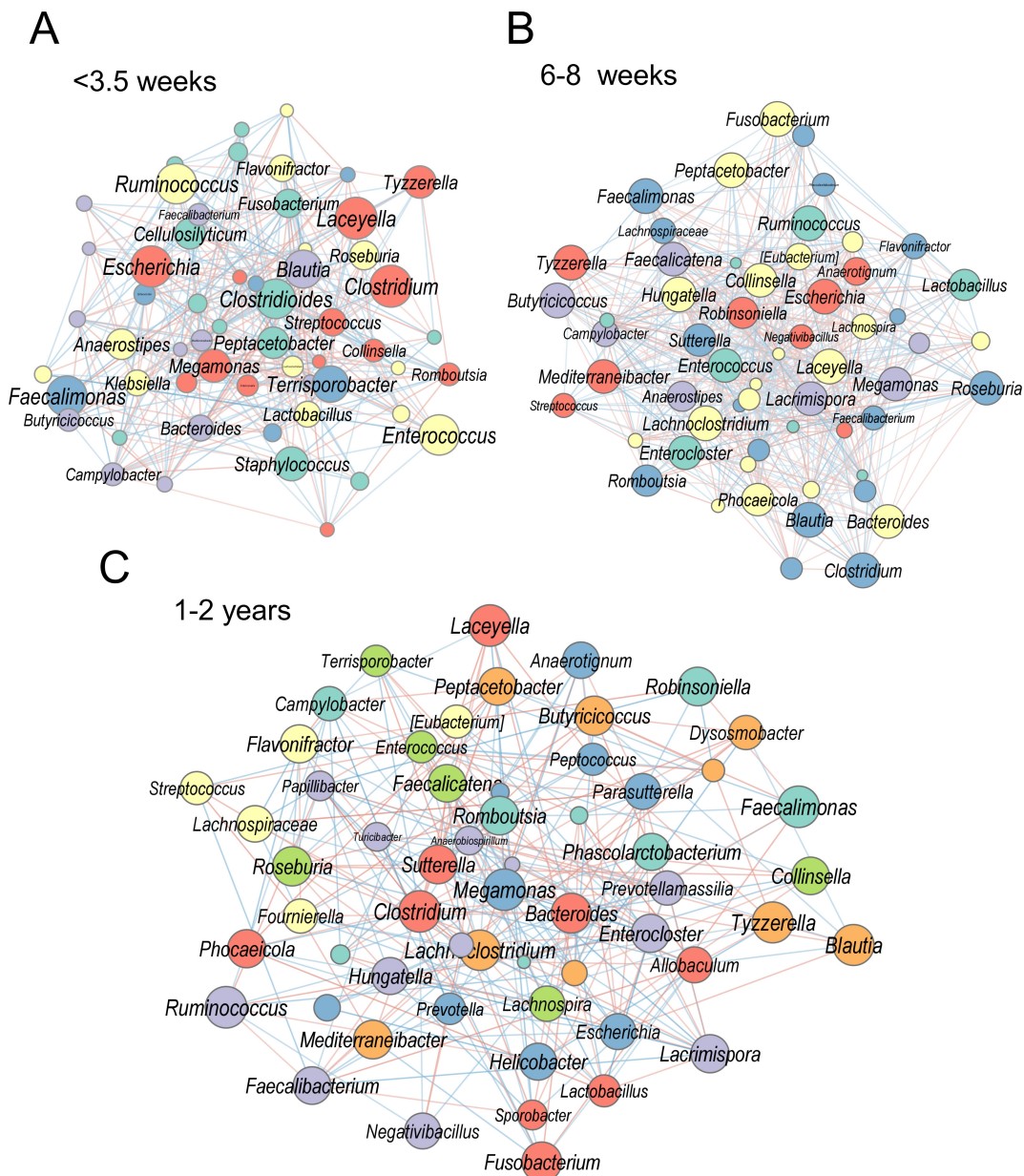

**FIG 4** Co-occurrence relationships between age categories reflect dietary shifts. (A) Co-occurrence network (SparCC) at up to 3.5 weeks. (B) Network at 6–8 weeks. (C) Network at 1–2 years. Node size is proportional to genus prevalence. Nodes are colored by Louvain cluster; edges are colored by correlation (blue = negative, red = positive).

rose steeply around weaning and plateaued by ~6 months, while ordination revealed an age-graded convergence in beta diversity. These patterns were consistent across kennels and litters, indicating that developmental dynamics are largely robust to kennel-level environmental heterogeneity.

Taxonomic analyses reinforced this trajectory. Early samples (0–8 weeks) were enriched for *Escherichia*, *Megamonas*, *Lactobacillus*, and *Ruminococcus*, whereas later stages showed dominance of *Blautia*, *Peptacetobacter*, and *Fusobacterium*, with *Blautia* increasing progressively across life stages. The short weaning window (6–8 weeks) emerged as a notable inflection point, including kennel-specific and diet-linked *Lactobacillus* patterns, consistent with a transition from milk-adapted to solid-food-adapted communities. Together, these findings emphasize weaning as a critical ecological breakpoint for canine gut community assembly.

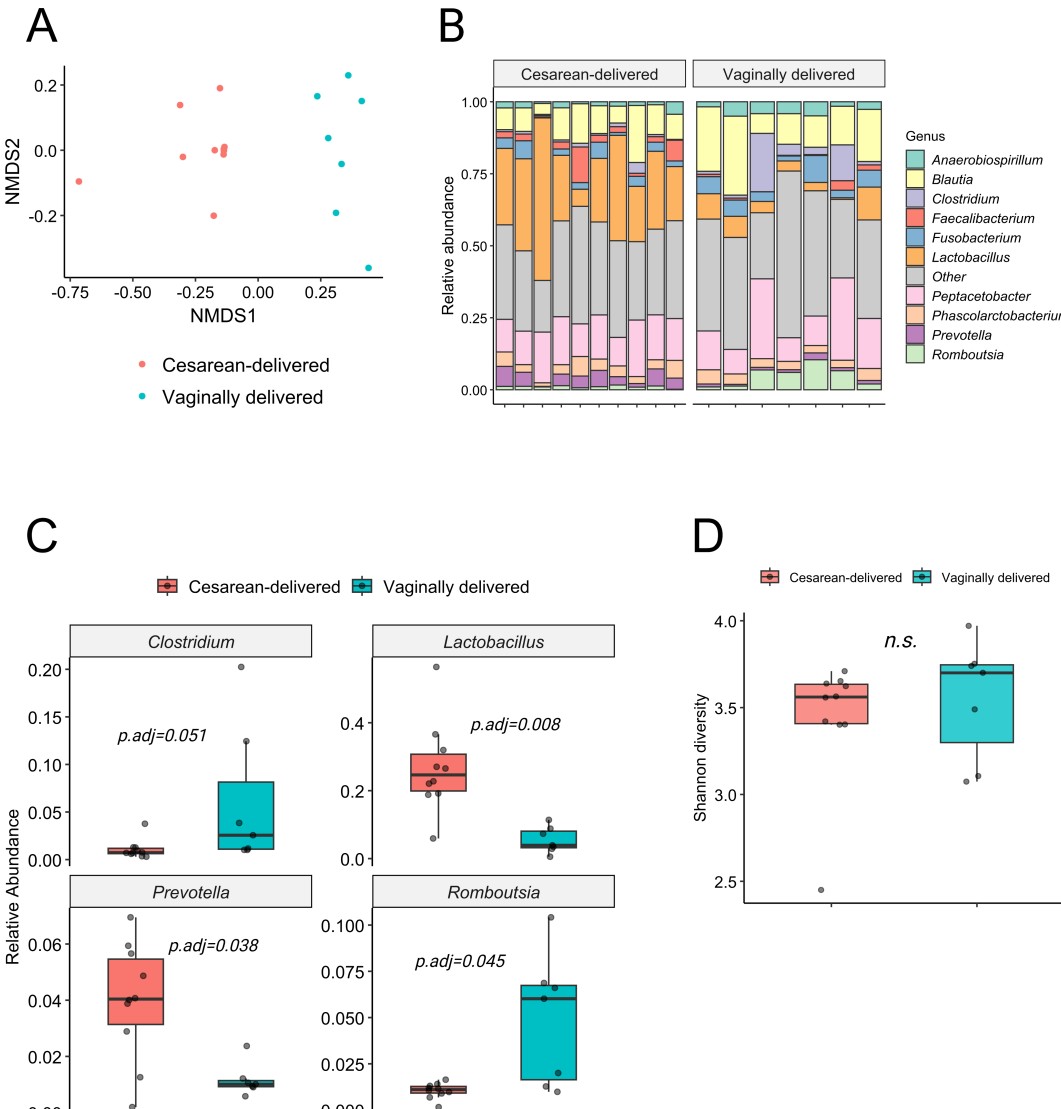

**FIG 5** Comparison of cesarean section vs. vaginally delivered pup microbiomes. (A) Non-metric multidimensional scaling based on delivery mode. (B) Shannon diversity between delivery modes. (C) Stacked bar plots of the relative abundances of the top 10 genera of puppies grouped by delivery mode. (D) Box plots of differential relative abundance according to the mode of delivery showing the top four significantly different genera (Wilcoxon rank-sum test, B-H adjusted *P*-values).

Comparable patterns have been described in humans and other mammals, where an early-life dominance of *Enterobacteriaceae* and *Lactobacillaceae* is progressively replaced by fiber-associated *Clostridia* (e.g., *Blautia*, *Ruminococcus*, *Faecalibacterium*) following the introduction of solid food (50). In carnivorous mammals, including dogs and cats, *Fusobacterium* often becomes a prominent post-weaning taxon, reflecting adaptation to meat-rich diets (15). The continuous increase of *Blautia* in our cohort parallels its rise in post-weaning communities across species and is consistent with its proposed role as a core genus during gut maturation (51). In contrast to humans, where *Bifidobacterium* can remain abundant during weaning (52), our dogs showed low and inconsistent *Bifidobacterium* levels throughout development. This pattern is consistent with previous canine studies reporting a lower abundance of this genus (15). Together, these similarities and differences underscore both the translational relevance of canine models and the species-specific features of microbiome development.

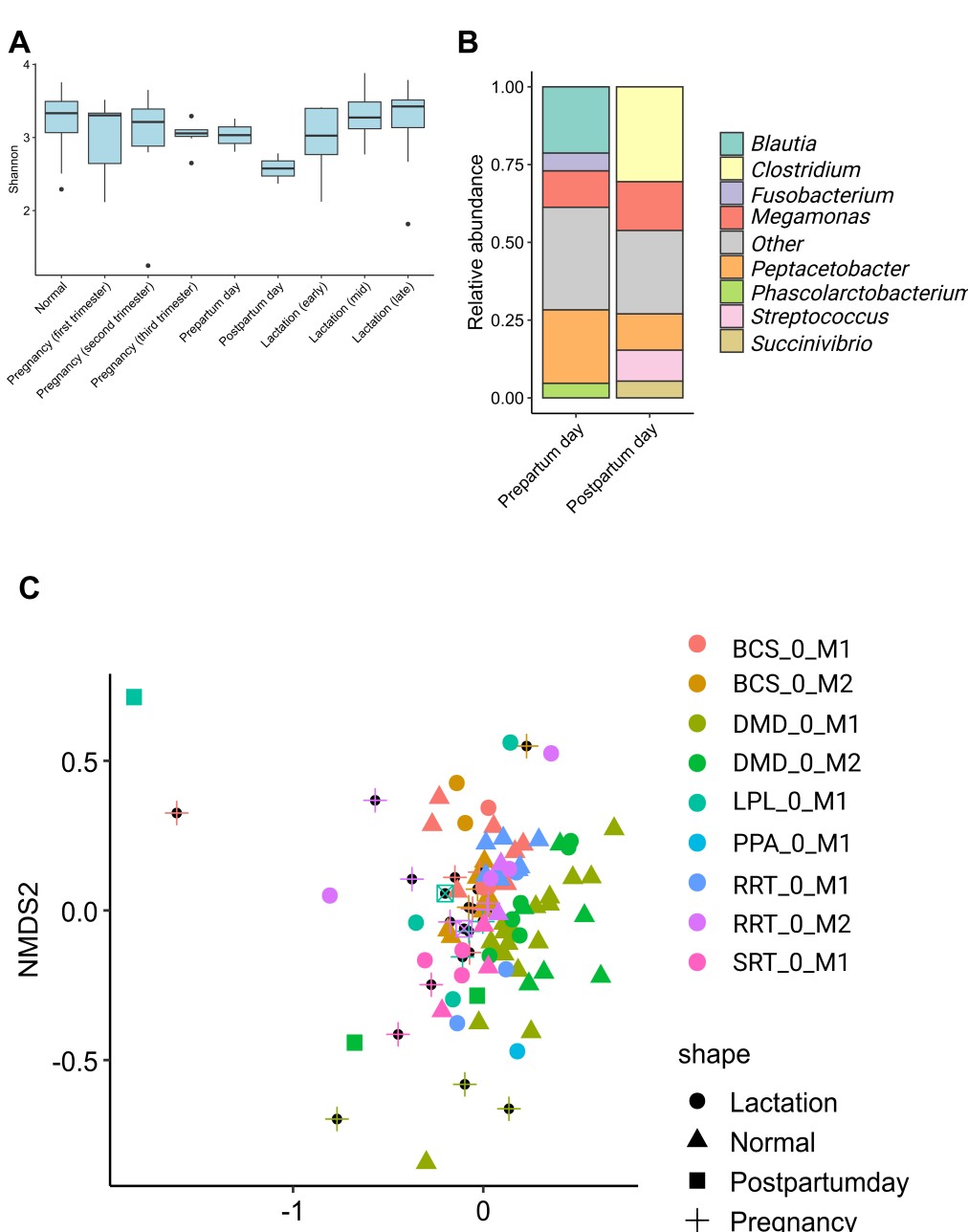

**FIG 6** Gut microbiome dynamics in mother dogs across reproductive states. (A) Distribution of alpha diversities within mother dogs throughout normal state, pregnancy, and lactation. (B) Stacked bar plots of relative abundances in the pre- and postpartum days show a temporary shift in the microbiome. (C) Unsupervised NMDS clustering reveals that microbiome profiles cluster more strongly by mother identity than by reproductive state, with no consistent association with kennel or diet.

Modeling of covariates demonstrated that age is the dominant predictor of alpha diversity, with kennel exerting a smaller yet detectable effect and host sex showing no measurable impact in this cohort. Comparisons between litters within the same kennel also revealed generally modest differences. These results suggest that, under typical husbandry and diet practices, intrinsic developmental programs and age-coupled diet transitions outweigh contextual environmental variation in shaping community structure. Practically, this implies that longitudinal age matching and careful accounting

for weaning status are essential for study design and interpretation in canine micro-biome research.

Network analyses indicated that microbe-microbe association structure also matures with age. SparCC-based association networks were sparse in neonates, became denser and more modular at 6–8 weeks, and were robust and highly connected in adults, with strong positive correlations among core taxa (e.g., *Bacteroides*, *Clostridium*, and *Fusobacterium*). While correlation does not establish interaction, the observed densification and modularity are consistent with increasing niche differentiation and cross-feeding potential as diets diversify and host physiology stabilizes.

Notably, C-section pups showed higher relative abundances of *Lactobacillus* (*P*.adj = 0.008) and *Prevotella* (*P*.adj = 0.038) and a lower abundance of *Romboutsia* (*P*.adj = 0.045) than vaginally delivered pups (Fig. 5C and D). At 8–10 weeks, no difference in Shannon diversity was observed between groups. This pattern diverges from the most common human neonatal reports for *Prevotella*, and from many studies reporting lower *Lactobacillus* in C-section neonates, yet is consistent with more recent data by Pahirah et al. (53), who found higher *Lactobacillus* in cesarean-delivered human neonates. Thus, delivery mode-related genus-level shifts are unlikely to be uniform across species or time and may depend on species-specific postnatal ecology (intense dam-pup contact, coprophagy), kennel environment, weaning practices, and the later sampling window in our study (8–10 weeks). Given that our C-section cohort comprised a single litter, these observations should be interpreted cautiously and validated in larger, multicenter cohorts that explicitly track perinatal practices. In neonatal dogs, Zakošek Pipan et al. (19) reported that vaginally delivered puppies exhibited meconium microbiota primarily of maternal vaginal origin, whereas cesarean-delivered puppies showed lower diversity with mixed resemblance to maternal vaginal and oral bacteria. Another study by Rota et al. (21) reported that in cesarean-delivered puppies, amniotic fluid and meconium were dominated by *Acinetobacter*, *Staphylococcus* spp., and *Bacillus* spp. as primary taxa. Together with our findings, these data suggest that while the existence of delivery mode-associated microbiota differences is conserved, the direction and taxon-level patterns are context- and time-dependent.

We also observed stage-linked shifts in dams, with transient reductions in alpha diversity during late pregnancy and early lactation and compositional turnover across the pre- to postpartum transition. These maternal dynamics likely influence early seeding and may interact with birth mode and nursing behaviors to shape early pup communities. Future work that jointly profiles dams and pups at higher temporal resolution, including milk and environmental sources, will help disentangle vertical and horizontal transmission routes.

Methodologically, this study leverages full-length 16S rRNA sequencing to achieve species-level resolution across a dense longitudinal design, minimizing batch effects and enabling fine-scale taxonomic and association analyses. Nonetheless, several limitations merit note. First, while kennel effects were modest, our cohort is breed-specific; generalization beyond Pumis requires replication across breeds and mixed-breed populations. Second, the observational design cannot establish causality, and co-occurrence networks derived from compositional data capture correlations rather than direct interactions. Third, amplicon sequencing limits functional inference; integrating shotgun metagenomics, metatranscriptomics, and metabolomics would clarify metabolic capacity and activity across developmental stages. Fourth, the cesarean comparison was underpowered (single litter), and residual confounding (e.g., subtle husbandry differences) cannot be fully excluded. Finally, it should be acknowledged that fecal microbiome profiling has limited ability to fully represent microbial communities along the entire gastrointestinal tract.

Taken together, our results position age—and the diet transitions that accompany it—as the primary drivers of canine gut microbiome assembly, with kennel/litter context and host sex exerting comparatively smaller effects. However, because age and diet co-vary in early life, their effects cannot be fully disentangled; by contrast, adult diets

were stable (BARF confined to a single kennel), limiting adult diet-related confounding. We identify weaning as a key window of rapid ecological change, document progressive remodeling of association network topology with development, and provide preliminary evidence that birth mode can shape early postnatal composition under otherwise controlled conditions. These insights support the use of dogs as a translational model for early-life microbiome development and highlight practical levers—age matching, weaning status, and perinatal factors—to improve study design and potential intervention strategies. Future studies that expand cesarean cohorts, incorporate functional multi-omics, and prospectively track dam-pup-environment triads will be critical to establishing mechanisms and guiding targeted microbiome modulation in veterinary and comparative medicine.

## MATERIALS AND METHODS

### Experimental animals and feeding protocols

This study included a main cohort of 84 healthy purebred Hungarian Pumi dogs comprising 55 puppies from 12 naturally (vaginally) delivered litters, their mothers, and additional adult male and female counterparts, all originating from six independent breeding kennels in Hungary that exclusively housed Hungarian Pumi dogs. Neonatal puppies were kept indoors in cradles, growing puppies in mixed indoor/garden environments, and adult dogs in similar house/garden areas, providing a consistent environmental background across kennels. Kennel abbreviations used throughout are as follows: BCS = Bükki Cserfes; DMD = Duna-menti Dumás; SRT = Serteperti; PPA = Pattogó Parázs; RRT = Rezerta-Réti; and LPL = Le Petit Lapin. Puppies were followed longitudinally from birth until adulthood, up to 81 weeks of age. The average age of the mothers was $4.3 \pm 1.6$ years (mean $\pm$ SD), while the control adult dogs had a mean age of $5.9 \pm 3.6$ years (mean $\pm$ SD) (see Table 1). Additionally, a subgroup of five puppies (10 samples) born by cesarean section from a single litter in the Le Petit Lapin kennel was included, delivered by the same dam that previously gave birth vaginally; C-section samples were analyzed separately and excluded from the primary cohort analysis presented in Table 1 (see Table 2 for details).

Puppies were exclusively breastfed until ~8 weeks, with solid food introduced at ~3.5 weeks alongside continued nursing. All dogs were primarily fed dry commercial diets; however, a subset of adult dogs from one kennel (Duna-menti Dumás) received a raw food diet (BARF). Diet composition and classification followed 2024 FEDIAF guidelines (54); details are provided in Table S1.

**TABLE 1** Study population summary by kennel, including litter delivery type, puppy sex distribution, maternal age, adult counterparts, and total sample counts

| Kennel ID | Number of litters | Number of puppies | | | Number of unique mother(s) | Mothers' age (years [mean ± SD]) | Number of adult female controls | Number of adult male controls | Control adults' age (years [mean ± SD]) | Samples per kennel |
|---|---|---|---|---|---|---|---|---|---|---|
| | | Total | Female | Male | | | | | | |
| BCS | 2 | 7 | 3 | 4 | 2 | 3.9 ± 1.6 | 2 | 0 | 3.2 ± 2.5 | 61 |
| DMD | 4 | 20 | 13 | 7 | 2[a] | 4.5 ± 3.2 | 0 | 1 | 6.8 | 114 |
| SRT | 2 | 4 | 1 | 3 | 1[b] | 4.2 | 3 | 4 | 6.2 ± 1.9 | 80 |
| PPA | 1 | 7 | 2 | 5 | 1 | 6.0 | 4 | 3 | 7.9 ± 4.6 | 46 |
| RRT | 2 | 12 | 5 | 7 | 2 | 4.0 ± 1.8 | 1 | 1 | 1.8 ± 0.1 | 100 |
| LPL | 1 | 5 | 3 | 2 | 1[c] | 4.1 | 0 | 1 | 1.8 | 45 |
| Total | 12 | 55 | 27 | 28 | 9 | 4.3 ± 1.6 | 10 | 10 | 5.9 ± 3.6 | 446 |

[a]Indicates that both mothers gave birth twice during the study period.
[b]Refers to a case where the second mother was not sampled during gestation/lactation and therefore excluded from the maternal age calculation.
[c]Denotes that this mother later delivered an additional litter by cesarean section, which is summarized separately in Table 2. The mean age of adult control dogs was calculated at the time of first sample collection. For mother dogs, age was calculated at the time of parturition. Kennel IDs: BCS, Bükki Cserfes; DMD, Duna-menti Dumás; LPL, Le Petit Lapin; PPA, Pattogó Parázs; RRT, Rezerta-Reti; and SRT, Serteperti.

**TABLE 2** Summary of the cesarean section (C-section)-delivered puppy litter from kennel Le Petit Lapin (LPL)$^a$

| Kennel ID | Number of litters | Number of puppies | | | Mother ID | Samples per puppy | Total samples |
|---|---|---|---|---|---|---|---|
| | | Total | Female | Male | | | |
| LPL | 1 | 5 | 0 | 5 | Same as LPL 1$^c$ | 2 | 10 |

$^a$This litter was delivered via cesarean section by the same mother who also delivered a vaginal litter included in Table 1 (see footnote c). These samples were analyzed separately and were excluded from the main cohort analysis presented in Table 1.

None of the dogs enrolled in this study received antibiotic treatment within 6 months prior to their first sampling time point or at any time during the sampling period; any animal requiring antibiotics would have been excluded from the cohort.

## Study samples

Stool samples were collected at multiple time points from dams during gestation, lactation, and post-lactation, from their offspring from birth through adulthood (up to 81 weeks), and from adult control. In total, 463 samples were collected and sequenced, of which 456 were included in the analyses presented here. The remaining seven samples (see Table S2 for accession numbers and exclusion criteria) are available in ENA (BioProject accession PRJEB82125) but were excluded from our analyses. These seven samples included one sample from a single puppy, not part of the main cohort, which accounts for the difference between the 90 dogs archived in ENA and the 89 dogs included in this study. All samples were obtained immediately after defecation using DNA/RNA Shield Fecal Collection Tubes (Zymo Research) and stored at −80°C until extraction.

## DNA extraction

Genomic DNA was extracted from stool samples using the ZymoBIOMICS 96 MagBead DNA Kit-Dx (Zymo Research, # Cat. No. D4308-E) following the manufacturer's protocol under sterile conditions in a Class II biosafety cabinet. Samples were homogenized before aliquoting, and bead-beating plus magnetic bead-based purification steps were performed according to the manufacturer's instructions. DNA yield (Table S3) and quality were assessed by a Qubit 4 fluorometer and Agilent 4150 TapeStation. gDNA was stored at −20°C until library preparation.

## 16S library preparation and sequencing

Full-length 16S libraries (V1–V9) were prepared from high-quality gDNA (DINs ≥6) using Oxford Nanopore Technologies' Rapid Sequencing Amplicons-16S Barcoding Kit (SQK-16S024) with AMPure XP bead cleanup following the manufacturer's protocol. Library concentration (Table S3) and fragment size (~1,500 bp) were verified using Qubit 4 and Agilent 4150 TapeStation before pooling, respectively. Libraries were pooled equimolarly and sequenced on an ONT MinION (Version R9; 25 flow cells total). Sequencing raw data were basecalled with Dorado v0.8.0 in super accurate mode (minimum Q score = 10) before downstream analysis. See Table S4 for details of the libraries pooled per flow cell and sequencing read statistics.

## Bioinformatic analysis

EMU v3.4.5 was used to generate read-count tables per taxon and calculate relative abundances following the workflow described by Curry et al. (55). Diversity metrics were computed from unfiltered counts using phyloseq v3.2 in R v4.2 (56). Taxa were filtered to those with >5% relative abundance and presence in ≥5 samples. Compositional differential abundance testing across age groups, diet categories, kennels, reproductive states, and delivery modes was performed using ANCOM-BC (ancombc R package) on the filtered count tables. The adjusted $P$-values ($P$.adj) reported in Results correspond

to ANCOM-BC outputs. Mixed-effects models (lme4) assessed the effects of age, sex, kennel, kennel:litter, and major diet categories; age was modeled both categorically and with splines to account for diet-age confounding. Major diet categories were defined as exclusive breast milk, mixed breast milk + starter kibble during the weaning period, post-weaning commercial kibble diets (17 formulations collapsed into macronutrient-based groups following FEDIAF protein:carbohydrate guidelines), and adult BARF diets.

A taxonomic tree of the top 350 taxa was visualized using iTOL (57), and genus-level hierarchical clustering and heatmaps were produced with ComplexHeatmap (58). Microbial co-occurrence networks were inferred with SparCC (SpiecEasi) and visualized using igraph and Cytoscape (59). Comparative statistics (Wilcoxon rank-sum, Student's $t$-test, Mann-Whitney $U$-test) were applied as indicated in figure legends, with $P < 0.05$ considered significant unless stated otherwise.

## Core microbiome definition and visualization

Core sets were defined at the species level using prevalence thresholds within each age group. A strict cutoff (≥90% prevalence) was applied for the heatmap and Jaccard similarity analyses, while a more inclusive cutoff (≥ 80% prevalence) was used for the UpSet plot, consistent with established practices in microbiome research (60). Heatmaps and Jaccard indices were computed from prevalence tables, and intersections among core sets were visualized using an UpSet plot. All visualizations were generated in Python (v. 3.12.3) using the pandas, numpy, and matplotlib libraries. The UpSet representation was created with custom plotting code based on matplotlib.

DNA isolation, library preparation, and the bioinformatic pipeline followed current best-practice guidelines as described in recent benchmarking studies (61, 62). Detailed methods are provided in File S1.

## ACKNOWLEDGMENTS

This work was supported by the Momentum Program I of the Hungarian Academy of Sciences LP2020-8/2020 (D.T.) and by the National Research, Development and Innovation Office (NRDIO) FK 142676 (D.T.). The APC fee was covered by the University of Szeged Open Access Fund: 8004. M.A., A.T., and N.J.Y. are awardees of the Stipendium Hungaricum Scholarship for full-time Ph.D. studies under registry numbers SHE-29026-004/2021, SHE-102901-004/2022, and SHE-33967-004/2023, respectively. The project was also funded by the NRDIO grant K 142674 to Z.B.

We would like to express our sincere gratitude to the following kennel owners for providing samples from their dogs (Table S3): Ildikó Abonyi, owner of the Duna-menti Dumás Kennel (Baja, Hungary, https://www.pumikennel.eu/), Dr. Csaba Dobó-Nagy, owner of the Pattogó Parázs Kennel (Gödöllő, Hungary, https://pumi.pedigreedatabaseonline.com/hu/Pattog%C3%B3-Par%C3%A1zs-kennel-Csaba-Dobo-Nagy/breeder/638), Viktória Hordósné Vasas, owner of the Bükki Cserfes Pumi Kennel (Bükkzsérc, Hungary, https://www.xn--bkki-cserfes-pumi-kennel-vsc.hu/), Gabriella Kassai, owner of the Serteperti Pumi Kennel (Kiskunmajsa [formerly Dány], Hungary, https://serteperti.hu/), Bianka Máthé-Szentes, owner of the Le Petit Lapin Pumi Kennel (Zomba, Hungary, https://pumi.mozellosite.com/), and Melinda Takácsné Horváth, owner of the Rezerta-Réti Pumi Kennel (Ajka-Padragkút, Hungary, https://www.facebook.com/profile.php?id=100057382122448). We would also like to thank the following dog owners for kindly providing samples: Kimberley Birkett (Beni; Finland), Eszter Dandé (Borcsa; Hungary), Dénes Gergely (Magor; Hungary), Tünde Gyöngyös (Briós; Switzerland), Ildikó Hernádi (Csúzli; Hungary), Yvett Király (Csepke; Hungary), Júlia Lang (Blöff; Hungary), Minna Silander (Pusta; Finland), and Éva Szőke (Rozi; Hungary). Finally, we are especially grateful to Tünde Baloghné Jancsovics, a board member of the Hungarian Pumi Club (https://pumiklub.eu/), for establishing the crucial contact with breeders and facilitating the collaboration. We gratefully acknowledge Balázs Kakuk (Department of Medical Biology [DMB], University of Szeged USZ]) for his work on

basecalling and EMU output generation. We also thank Zsolt Csabai (DMB, USZ) and Ákos Dörmő (DMB, USZ) for laboratory support and helpful discussions.

Conceptualization: D.T. and Z.B. Methodology: D.T., Z.B., P.O., and M.A. Investigation: M.A., A.T., N.J.Y., T.J., and G.G. Data curation: M.A., T.J., and G.G. Formal analysis: P.O., D.T., and M.A. Validation: D.T. and M.A. Project administration: D.T. Supervision: D.T. Writing—original draft: D.T., M.A., A.T., P.O., and Z.B. Writing—review and editing: M.A., A.T., N.J.Y., T.J., G.G., P.O., D.T., and Z.B. All authors read and approved the final paper.

## AUTHOR AFFILIATIONS

[1]MTA-SZTE Lendület GeMiNI Research Group, University of Szeged, Szeged, Hungary
[2]Department of Medical Biology, Albert Szent-Györgyi Medical School, University of Szeged, Szeged, Hungary
[3]Department of Dermatology, Medical Faculty, University Hospital Duesseldorf, Heinrich-Heine University Duesseldorf, Duesseldorf, Germany

## AUTHOR ORCIDs

Zsolt Boldogkői ⓘ https://orcid.org/0000-0003-1184-7293
Dóra Tombácz ⓘ http://orcid.org/0000-0001-5520-2978

## FUNDING

| Funder | Grant(s) | Author(s) |
| --- | --- | --- |
| Magyar Tudományos Akadémia | Lendület LP2020-8/2020 | Dóra Tombácz |
| Nemzeti Kutatási Fejlesztési és Innovációs Hivatal | FK 142676 | Dóra Tombácz |
| Nemzeti Kutatási Fejlesztési és Innovációs Hivatal | K 142674 | Zsolt Boldogkői |

## AUTHOR CONTRIBUTIONS

Md Asaduzzaman, Data curation, Formal analysis, Investigation, Methodology, Writing – original draft, Writing – review and editing | Péter Oláh, Formal analysis, Methodology, Validation, Visualization, Writing – original draft, Writing – review and editing | Natheer Jameel Yaseen, Investigation, Writing – review and editing | Ahmed Taifi, Investigation, Writing – original draft, Writing – review and editing | Tamás Járay, Data curation, Investigation, Writing – review and editing | Gábor Gulyás, Data curation, Investigation, Writing – review and editing | Zsolt Boldogkői, Conceptualization, Methodology, Resources, Writing – original draft, Writing – review and editing | Dóra Tombácz, Conceptualization, Data curation, Formal analysis, Funding acquisition, Methodology, Project administration, Resources, Supervision, Validation, Visualization, Writing – original draft, Writing – review and editing

## DATA AVAILABILITY

The sequencing raw data were basecalled using Dorado in super-accurate mode (Q ≥ 10), and the resulting FASTQ files have been deposited in the European Nucleotide Archive (ENA) under the BioProject accession number PRJEB82125. The repository also contains sample-associated metadata, and the metadata used for analyses are provided in Tables S5 and S6.

## ETHICS APPROVAL

In accordance with institutional and national regulations, formal animal ethics approval was not required for this work. The study involved only noninvasive fecal sampling during routine husbandry without any changes to housing, diet, or veterinary care;

samples were provided voluntarily by breeders. All procedures complied with national animal welfare regulations.

## ADDITIONAL FILES

The following material is available online.

### Supplemental Material

**Supplemental Figures (mSystems01279-25-s0001.pdf).** Figures S1 to S7.
**Supplementary File (mSystems01279-25-s0002.pdf).** Detailed laboratory and bioinformatic protocols, including DNA extraction, 16S rRNA library preparation, sequencing, and data processing steps.
**Supplemental Tables (mSystems01279-25-s0003.xlsx).** Tables S1 to S6.

### Open Peer Review

**PEER REVIEW HISTORY (review-history.pdf).** An accounting of the reviewer comments and feedback.

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
