## [Reviewer comments · mSystems]

Longitudinal Long-Read Microbiome Profiling in a Canine Model Reveals How Age, Diet, and Birth Mode Shape Gut Community Dynamics

Md Asaduzzaman, Péter Oláh, Natheer Yaseen, Ahmed Taifi, Tamás Járny, Gábor Gulyás, Zsolt Boldogkői, and Dóra Tombácz

Corresponding Author(s): Dóra Tombácz, Szegedi Tudományegyetem

Review Timeline:

Submission Date:	September 2, 2025
Editorial Decision:	October 17, 2025
Revision Received:	November 19, 2025
Accepted:	December 16, 2025

Editor: Tricia Van Laar

Reviewer(s): The reviewers have opted to remain anonymous.

Transaction Report:

DOI: <https://doi.org/10.1128/msystems.01279-25>

Re: mSystems01279-25 (Longitudinal Long-Read Microbiome Profiling in a Canine Model Reveals How Age, Diet, and Birth Mode Shape Gut Community Dynamics)

Dear Dr. Dóra Tombácz:

We enjoyed your study very much and only have minor concerns. Those are included below. When I tried to access the BioProject on NCBI, for some reason the BioProject itself doesn't work, but I was able to find the individual sample reads. This is likely due to a minor issue on NCBI's part, but with the current US government shutdown, it may not be something that can be addressed right away. In any case, I just wanted to let you know.

Revision Guidelines

Sincerely,
Tricia Van Laar
Editor
mSystems

Reviewer #1 (Comments for the Author):

This elegant and thorough study presents a longitudinal assessment of the changes in the faecal microbiome of a single breed

of dogs overtime, including assessment of dams and covering birth, through to weaning and adulthood. The findings are eloquently discussed and I found the manuscript well written and interesting. It should become the benchmark for future studies testing the results of interventions given that the study has used complete 16S rDNA profiles enabling identification to the species level and has been well designed and executed. The uniformity of microbiome profiles attests to the excellent study design.

The study identified that age and diet were more influential on faecal microbiome structure compared to sex and 13 different species were present in age groups. Weaning was identified as the major ecological breakpoint and interestingly, differences in microbiome composition were demonstrated between caesarean and vaginally delivered pups with important genera such as *Lactobacillus* having higher relative abundance in the former group. The authors attest that clearly more studies need to be undertaken in this subgroup and displayed the right level of circumspection. Finally, the gut microbiota of dams also produced significant findings associated with reproductive status, also a timely addition to the literature.

Specific comments are found below

Line 48 Space between end of sentences and references.

Line 191-193: This is interpretation in the results section and needs to be removed, it is well covered in the Discussion.

Line 225: Is this limited prevalence of *Bifidobacterium*, or limited abundance, or both? Unclear from the way it is written.

Line 243: It appears contradictory to outline that the findings of increased relative abundance of *Lactobacillus* and *Prevotella* in caesarean vs vaginal delivered pups are at odds with the findings in human literature with the referenced statement in lines 253-256 which shows agreement. Please explain.

Study limitations: Very pleasing to see such a robust group of study limitations. Is it possible that one more should be added in that analysing the faecal microbiome may not be representative of different compartments within the gut?

Ethics approval: Not required given that passed faecal samples were collected.

Reviewer #2 (Comments for the Author):

Wonderful study! I made several comments, but all are minor suggestions. See below and attached document.

Review: "Longitudinal Long-Read Microbiome Profiling in a Canine Model Reveals How Age, Diet, and Birth Mode Shape Gut Community Dynamics"

I would like to thank the authors for submitting this manuscript. This provides important insight into the development and composition of the canine microbiome. The manuscript was well-written and informative. Please see individual comments below.

Introduction:

Line 59-61: I am sure you are aware that there remains controversy in this area. There is a commentary published in *Microbiome* called "Lessons learned from the prenatal microbiome controversy", which offers some nice summary perspectives. While I do not think there are any major issues with what you said here, you could consider acknowledging the controversy since some readers might have strong feelings on this. This is a minor suggestion, and I also think it's fine if you leave it as is.

Line 65: While I agree that studies on canine delivery method are scarce, I suggest citing those available since there are some addressing this.

Line 91: The term "naturally reared" can have different implications across different groups and individuals. For instance, some consider this to mean only feeding raw diets, minimal-no vaccinations, strict chemical avoidance, etc. Since this is not a term that you define elsewhere, I actually would suggest not using it here, as it might create some confusion or misinterpretation. I would consider either removing this or using more specific wording.

Results:

Line 96: Can you include the number of dams and time-matched controls here?

Line 100: Typo (missing space) after ">5%,"

Methods:

Line 290: I remain a bit unclear as to what these kennels are. Are they breeding kennels, working dog kennels, research colonies; do they have other dogs or even other animal species around, etc? You talk about generalization of the findings to other breeds in line 266 of the discussion. Understanding a bit about their lifestyle/environment might also apply here. For instance, in addition to breed differences, biome findings in dogs at a breeding kennel might look very different from a more controlled research colony, and this info could inform how translatable these findings are to dogs in homes or other

environments. Could you add a line about what types of kennels these are/the environment?

Table S1: I appreciate the inclusion of detailed diet information in the supplemental. This is lovely. You could consider adding the caloric density (kcal/kg) for all diets so that people can convert nutrient % to g/100 kcal for direct comparisons. You currently do this for some but not all diets.

General comment: If possible to add additional details on age-diet interactions, that would be helpful. The only details I find regarding diet shifts over time are the comments on line 298-299, stating that puppies were exclusively breastfed until ~8 weeks, with solid food introduced at ~3.5 weeks alongside continued nursing. Do you know the age at which dogs (or even most dogs) were fully transitioned onto their commercial diets? Were diets continuing to change, or were most diets stable after the initial transition from nursing to commercial diet? Also, you include a mixed effects model that includes age, kennel, diet, and sex, but in figure S5, it appears that the only diet analysis was specifically for BARF-diet. In figure 2C and in the analyses, it is unclear what the diet effect size is measuring. Is this only the effects of raw, or is this breastmilk vs kibble, or is it comparing individual kibble diets based on nutrient profiles? The methods just say "major diet categories". Since many of the age-related shifts correspond to the times of diet transitions, I think you should add some to clarify these points. I appreciate your discussion about these limitations (e.g., line 275-277).

Line 301-302: For antibiotics, not receiving antibiotics during sampling is great, but it raises the possibility that animals could have had antibiotics shortly before sampling where the effects are still present. Do you have that info? Could you say if any animals had antibiotics within X months prior to sampling or provide any additional details here?

Lines 329: Include how you performed differential abundance testing in this section (e.g., ANCOM, other).

Figures:

Overall nice figures! I have a few comments/suggestions described below.

Figure 2B: The text along the top is cut off (e.g. "All_Ag2_breastmilk_complement" reads "g2_breastmilk_complme").

Figure 2D: Text legend and the text in the results section say there is one significant difference, but there's no asterisk or indicator to show which one. I can guess based on the figure, but you could add an asterisk to highlight.

Figure 3A: The figure legend says this is stratified by kennel, litter, and life phase in the text, yet the barplot appears to only be divided by life phase/age.

Figure 3B: Is this showing taxa from across all samples (all kennels, litters, ages, etc)? I would clarify in the legend. Also, a minor note, but I cannot tell if the smaller dots in the lower right are black (representing a different genus), or if these are blue and still represent *Blautia*, just appearing darker because of the density. Lastly, for genera in both A and B, it would be great if they could have the same colors in both figures to avoid confusion. For instance, *P. hiranonis* is a blue/purple color in panel B, but *peptacetobacter* is green in A. If possible to make these consistent, it would improve clarity. Not a major comment if this is not possible.

Figure 3C: If space, one suggestion is to add a key for the kennel colors as well. Again, this is certainly not a major issue, so okay if not feasible. I can imagine a couple scenarios where that might be helpful for the reader.

Figure 6: Is it possible to include an NMDS showing the individual mothers in different colors, with different shapes (or whatever you like) indicating the reproductive state (e.g., normal, first trimester, lactation, etc). This would help clarify how many different mothers were sampled, which is currently unclear. Also, it would show how much variation there is within vs between individuals, which is great information. For instance, even as an individual mother shows these shifts throughout reproductive states, does her microbiome stay more similar to herself throughout that process, or in contrast, does she cluster more closely with another mother in the same reproductive state. My guess is the former, but a simple figure could provide great insight here.

Review: “**Longitudinal Long-Read Microbiome Profiling in a Canine Model Reveals How Age, Diet, and Birth Mode Shape Gut Community Dynamics**”

I would like to thank the authors for submitting this manuscript. This provides important insight into the development and composition of the canine microbiome. The manuscript was well-written and informative. Please see individual comments below.

Introduction:

Line 59-61: I am sure you are aware that there remains controversy in this area. There is a commentary published in *Microbiome* called “Lessons learned from the prenatal microbiome controversy”, which offers some nice summary perspectives. While I do not think there are any major issues with what you said here, you could consider acknowledging the controversy since some readers might have strong feelings on this. This is a minor suggestion, and I also think it’s fine if you leave it as is.

Line 65: While I agree that studies on canine delivery method are scarce, I suggest citing those available since there are some addressing this.

Line 91: The term “naturally reared” can have different implications across different groups and individuals. For instance, some consider this to mean only feeding raw diets, minimal-no vaccinations, strict chemical avoidance, etc. Since this is not a term that you define elsewhere, I actually would suggest not using it here, as it might create some confusion or misinterpretation. I would consider either removing this or using more specific wording.

Results:

Line 96: Can you include the number of dams and time-matched controls here?

Line 100: Typo (missing space) after “(>5%;

Methods:

Line 290: I remain a bit unclear as to what these kennels are. Are they breeding kennels, working dog kennels, research colonies; do they have other dogs or even other animal species around, etc? You talk about generalization of the findings to other breeds in line 266 of the discussion. Understanding a bit about their lifestyle/environment might also apply here. For instance, in addition to breed differences, biome findings in dogs at a breeding kennel might look very different from a more controlled research colony, and this info could inform how translatable these findings are to dogs in homes or other environments. Could you add a line about what types of kennels these are/the environment?

Table S1: I appreciate the inclusion of detailed diet information in the supplemental. This is lovely. You could consider adding the caloric density (kcal/kg) for all diets so that people can convert nutrient % to g/100 kcal for direct comparisons. You currently do this for some but not all diets.

General comment: If possible to add additional details on age-diet interactions, that would be helpful. The only details I find regarding diet shifts over time are the comments on line 298-299, stating that puppies were exclusively breastfed until ~8 weeks, with solid food introduced at ~3.5 weeks alongside continued nursing. Do you know the age at which dogs (or even most dogs) were fully transitioned onto their commercial diets? Were diets continuing to change, or were most diets stable after the initial transition from nursing to commercial diet? Also, you include a mixed effects model that includes age, kennel, diet, and sex, but in figure S5, it appears that the only diet analysis was specifically for BARF-diet. In figure 2C and in the analyses, it is unclear what the diet effect size is measuring. Is this only the effects of raw, or is this breastmilk vs kibble, or is it comparing individual kibble diets based on nutrient profiles? The methods just say “major diet categories”. Since many of the age-related shifts correspond to the times of diet transitions, I think you should add some to clarify these points. I appreciate your discussion about these limitations (e.g., line 275-277).

Line 301-302: For antibiotics, not receiving antibiotics during sampling is great, but it raises the possibility that animals could have had antibiotics shortly before sampling where the effects are still present. Do you have that info? Could you say if any animals had antibiotics within X months prior to sampling or provide any additional details here?

Lines 329: Include how you performed differential abundance testing in this section (e.g., ANCOM, other).

Figures:

Overall nice figures! I have a few comments/suggestions described below.

Figure 2B: The text along the top is cut off (e.g. “All_Ag2_breastmilk_complement” reads “g2_breastmilk_complme”).

Figure 2D: Text legend and the text in the results section say there is one significant difference, but there’s no asterisk or indicator to show which one. I can guess based on the figure, but you could add an asterisk to highlight.

Figure 3A: The figure legend says this is stratified by kennel, litter, and life phase in the text, yet the barplot appears to only be divided by life phase/age.

Figure 3B: Is this showing taxa from across all samples (all kennels, litters, ages, etc)? I would clarify in the legend. Also, a minor note, but I cannot tell if the smaller dots in the lower right are black (representing a different genus), or if these are blue and still represent *Blautia*, just appearing darker because of the density. Lastly, for genera in both A and B, it would be great if they could have the same colors in both figures to avoid confusion. For instance, *P. hiranonis* is a blue/purple color in panel B, but *peptacetobacter* is green in A. If possible to make these consistent, it would improve clarity. Not a major comment if this is not possible.

Figure 3C: If space, one suggestion is to add a key for the kennel colors as well. Again, this is certainly not a major issue, so okay if not feasible. I can imagine a couple scenarios where that might be helpful for the reader.

Figure 6: Is it possible to include an NMDS showing the individual mothers in different colors, with different shapes (or whatever you like) indicating the reproductive state (e.g., normal, first trimester, lactation, etc). This would help clarify how many different mothers were sampled, which is currently unclear. Also, it would show how much variation there is within vs between individuals, which is great information. For instance, even as an individual mother shows these shifts throughout reproductive states, does her microbiome stay more similar to herself throughout that process, or in contrast, does she cluster more closely with another mother in the same reproductive state. My guess is the former, but a simple figure could provide great insight here.

Reviewer 1

This elegant and thorough study presents a longitudinal assessment of the changes in the faecal microbiome of a single breed of dogs overtime, including assessment of dams and covering birth, through to weaning and adulthood. The findings are eloquently discussed and I found the manuscript well written and interesting. It should become the benchmark for future studies testing the results of interventions given that the study has used complete 16S rDNA profiles enabling identification to the species level and has been well designed and executed. The uniformity of microbiome profiles attests to the excellent study design. The study identified that age and diet were more influential on faecal microbiome structure compared to sex and 13 different species were present in age groups. Weaning was identified as the major ecological breakpoint and interestingly, differences in microbiome composition were demonstrated between caesarean and vaginally delivered pups with important genera such as Lactobacillus having higher relative abundance in the former group. The authors attest that clearly more studies need to be undertaken in this subgroup and displayed the right level of circumspection. Finally, the gut microbiota of dams also produced significant findings associated with reproductive status, also a timely addition to the literature.

We sincerely thank the reviewer for these exceptionally kind and encouraging comments. We are grateful for the positive assessment of the study design, analyses, and interpretation, and we truly appreciate the recognition of the work's breadth—from neonatal development to dam microbiota. Your thoughtful feedback is highly motivating and greatly valued.

Specific comments are found below

Line 48 Space between end of sentences and references.

This was a typographical error; the spacing has been corrected as requested.

Line 191-193: This is interpretation in the results section and needs to be removed; it is well covered in the Discussion.

The sentences in lines 191–193 have been removed from the Results section.

Line 225: Is this limited prevalence of Bifidobacterium, or limited abundance, or both? Unclear from the way it is written.

Previous studies have reported a lower relative abundance of *Bifidobacterium* compared to other major taxa in puppies. We have revised the sentence accordingly to clarify this point.

Line 243: It appears contradictory to outline that the findings of increased relative abundance of Lactobacillus and Prevotella in caesarean vs vaginal delivered pups are at odds with the findings in human literature with the referenced statement in lines 253-256 which shows agreement. Please explain.

We thank the reviewer for pointing out this apparent inconsistency. Our intention was to distinguish between two levels: (i) the general concept that delivery mode can shape early-life gut microbiota, and (ii) the direction of changes in specific genera. In our canine cohort, C-section pups at 8–10 weeks showed higher *Lactobacillus* and *Prevotella* and lower *Romboutsia* compared with vaginally delivered pups. This pattern diverges from many human neonatal studies reporting reduced *Bacteroides/Prevotella* and lower *Lactobacillus* in C-section infants, but is consistent with more recent human data (Pahirah et al., 2024) showing higher *Lactobacillus* in cesarean-delivered neonates. We have revised the Discussion to explicitly state that our results are partly consistent and partly divergent with the human literature and to suggest plausible explanations, including species-specific postnatal ecology (dam–pup contact, coprophagy), kennel environment, weaning practices, and our later sampling window (8–10 weeks). The revised text clarifies this point and avoids the impression of a logical contradiction.

Study limitations: Very pleasing to see such a robust group of study limitations. Is it possible that one more should be added in that analysing the faecal microbiome may not be representative of different compartments within the gut?

We agree that analysing the faecal microbiome may not fully represent microbial communities across all gut compartments. A brief statement acknowledging this has been added to the Discussion section among the study limitations.

Ethics approval: Not required given that passed faecal samples were collected

We confirm that ethics approval was not required, as only non-invasive, passively collected faecal samples were used. The Ethics section has been retained to clarify this point.

Reviewer 2

Wonderful study! I made several comments, but all are minor suggestions. See below and attached document.

Review: "Longitudinal Long-Read Microbiome Profiling in a Canine Model Reveals How Age, Diet, and Birth Mode Shape Gut Community Dynamics"

I would like to thank the authors for submitting this manuscript. This provides important insight into the development and composition of the canine microbiome. The manuscript was well-written and informative. Please see individual comments below.

Thank you very much for your kind assessment and constructive suggestions. We truly appreciate your thoughtful review and are grateful for the positive feedback on the clarity and value of our work.

Introduction:

Line 59-61: I am sure you are aware that there remains controversy in this area. There is a commentary published in Microbiome called "Lessons learned from the prenatal microbiome controversy", which offers some nice summary perspectives. While I do not think there are any major issues with what you said here, you could consider acknowledging the controversy since some readers might have strong feelings on this.

This is a minor suggestion, and I also think it's fine if you leave it as is.

We thank the reviewer for this thoughtful suggestion. We agree that the existence and extent of a prenatal microbiome remain debated. In the revised manuscript, we have added a clarifying sentence to explicitly acknowledge this controversy, noting that several authors have argued that many low-biomass signals may reflect contamination rather than stable fetal colonization (now lines 61–63). This change makes our position more balanced while remaining consistent with the current literature.

Line 65: While I agree that studies on canine delivery method are scarce, I suggest citing those available since there are some addressing this.

We agree that, while canine studies on delivery mode are limited, several recent reports have begun to address this question. We have revised the Introduction to explicitly cite these studies and to clarify that, although birth mode–associated differences in neonatal microbiota have been described in dogs, longitudinal and large-cohort datasets remain scarce.

Line 91: The term "naturally reared" can have different implications across different groups and individuals. For instance, some consider this to mean only feeding raw diets, minimal-no vaccinations, strict chemical avoidance, etc. Since this is not a term that you define elsewhere, I actually would suggest not using it here, as it might create some confusion or misinterpretation. I would consider either removing this or using more specific wording.

We agree that the term “naturally reared” can carry unintended interpretations in different contexts. To avoid any ambiguity, we have removed this phrase. The revised sentence now reads: “This work aims to advance our understanding of postnatal microbiome assembly in a translational model species.”

Results:

Line 96: Can you include the number of dams and time-matched controls here?

We agree that it is more appropriate to present this information clearly in the Results section. In our study, a total of nine mothers were included, which together produced twelve vaginally delivered litters (some mothers gave birth more than once during the study period; see Table 1 for details). Additionally, one of these bitches later gave birth to a litter by caesarean section, which was analysed separately (see Table 2 for clarification). Time-matched adult controls included ten females and ten males. All of this information was already presented in Tables 1 and 2 of the original manuscript; however, the column headers for female and male controls were previously mislabelled as “counterparts”. In the revised version, we have corrected the headers to “female controls” and “male controls”, and we have added the number of dams and time-matched adult controls in line 96, as suggested.

Line 100: Typo (missing space) after ">5%,"

The missing space has been inserted.

Methods:

Line 290: I remain a bit unclear as to what these kennels are. Are they breeding kennels, working dog kennels, research colonies; do they have other dogs or even other animal species around, etc.? You talk about generalization of the findings to other reeds in line 266 of the discussion. Understanding a bit about their lifestyle/environment might also apply here. For instance, in addition to breed differences, biome findings in dogs at a breeding kennel might look very different from a more controlled research colony, and this info could inform how translatable these findings are to dogs in homes or other environments. Could you add a line about what types of kennels these are/the environment?

All dogs in this study originated from breeding kennels (purebred Hungarian Pumi kennels) that housed only Hungarian Pumi dogs, with no other dog breeds or animal species present. All breeders are registered members of the Hungarian Pumi Club. In response to the reviewer's suggestion, we have now clarified the kennel type and living environment in the revised Methods section. Specifically, we state that neonatal puppies were kept in cradles indoors, growing puppies in combined indoor/garden areas, and adult dogs in similar house/garden environments, providing a consistent environmental background across kennels.

Metadata describing the living environment, collection site, and ambient temperature were collected for all samples and are publicly available in ENA BioProject PRJEB82125. In our analyses, age, diet, and kennel were the most influential variables; environmental factors within kennels did not show meaningful variation. We have updated the manuscript accordingly.

Table S1: I appreciate the inclusion of detailed diet information in the supplemental. This is lovely. You could consider adding the caloric density (kcal/kg) for all diets so that people can convert nutrient % to g/100 kcal for direct comparisons. You currently do this for some but not all diets.

Energy content (kcal/kg) has been added as a new column in Table S1 for all diets.

*General comment: If possible to add additional details on age-diet interactions, that would be helpful. The only details I find regarding diet shifts over time are the comments on line 298-299, stating that puppies were exclusively breastfed until ~8 weeks, with solid food introduced at ~3.5 weeks alongside continued nursing. Do you know the age at which dogs (or even most dogs) were fully transitioned onto their commercial diets? Were diets continuing to change, or were most diets stable after the initial transition from nursing to commercial diet? Also, you include a mixed effects model that includes age, kennel, diet, and sex, but in **figure S5**, it appears that the only diet analysis was specifically for BARF-diet.*

In our cohort, puppies began consuming solid food alongside breast milk at approximately 3.5 weeks of age. By 8 weeks of age, all puppies were fully weaned and transitioned to commercial diets; after this time point, diets remained stable within each kennel and did not continue to change during development. Thus, in the mixed-effects model, “diet” reflects the type of post-weaning diet (commercial vs. BARF), rather than ongoing dietary transitions.

Regarding Figure S5, our analysis proceeded in two steps. In Figure S5A, we specifically examined the effect of the BARF diet because this diet differs fundamentally in macronutrient composition and processing from commercial diets, and only one kennel consistently used BARF. We therefore evaluated BARF separately to avoid conflating its effects with commercial diet variation. In Figure S5B, we extended the analysis by assessing how the taxa identified as diet-responsive in S5A varied across diet types more broadly. This two-step approach allowed us to characterise both the specific impact of BARF and the broader patterns across commercial diet categories. We have clarified these points in the revised manuscript.

*In **figure 2C** and in the analyses, it is unclear what the diet effect size is measuring. Is this only the effects of raw, or is this breastmilk vs kibble, or is it comparing individual kibble diets based on nutrient profiles?*

We fully agree that clarification is needed. In our mixed-effects model, “diet” was included as a categorical variable representing the major diet types present in the dataset, rather than the exact nutrient composition of each food item. Specifically, the “diet” factor comprised:

- breastfeeding only (0–3.5 weeks),
- mixed breastfeeding + solid food (3.5–8 weeks),
- commercial kibble (after full transition at ~8 weeks),

- BARF diet (only in one kennel, affecting adult dogs).

Thus, the reported diet effect size reflects the overall variance in microbiome composition attributable to diet category, in comparison with age, kennel, and sex. It does not represent nutrient-level differences between individual commercial kibble brands. We have now clarified this in the Methods and Results to avoid ambiguity.

The methods just say "major diet categories". Since many of the age-related shifts correspond to the times of diet transitions, I think you should add some to clarify these points. I appreciate your discussion about these limitations (e.g., line 275-277).

In our models, “major diet categories” refer to biologically meaningful feeding types rather than individual commercial brands. Across the cohort, 17 different commercial kibble formulations were used. For the analyses, each sampling time point was assigned to one of the following categories: (i) exclusive breast milk (pre-weaning), (ii) mixed breast milk + starter kibble during the weaning window, (iii) post-weaning commercial kibble diets (17 formulations collapsed into macronutrient-based groups according to FEDIAF protein: carbohydrate recommendations), and (iv) raw BARF diet in a subset of adult dogs from a single kennel. In this sense, the “major diet” categories cover all diets present in the study, but they are modelled as coarse feeding types rather than fine-grained nutrient profiles. As noted in the Discussion (lines 275–277), age and diet transitions are tightly coupled in early life, so these effects cannot be fully disentangled in our dataset.

Line 301-302: For antibiotics, not receiving antibiotics during sampling is great, but it raises the possibility that animals could have had antibiotics shortly before sampling where the effects are still present. Do you have that info? Could you say if any animals had antibiotics within X months prior to sampling or provide any additional details here?

Prior to initiating the study, we obtained consent from all breeders and selected dogs based on good general health and body condition. We also systematically recorded recent medication history. None of the dogs enrolled in this study received any antibiotic treatment within six months prior to their first sampling time point, nor at any time during the sampling period. If a dog had required antibiotic therapy, it would have been excluded from this cohort and reserved for other projects. Thus, no samples included in the present analyses were collected from animals

exposed to antibiotics in the preceding six months or during follow-up. We have clarified this information in the revised version of the manuscript.

Lines 329: Include how you performed differential abundance testing in this section (e.g., ANCOM, other).

We have now explicitly described our approach to differential abundance testing in the Bioinformatic Analysis section. Briefly, compositional differential abundance analyses across age groups, diet categories, kennels, reproductive states, and delivery modes were performed using ANCOM-BC (ancombc R package) on filtered count tables, and the adjusted p-values (p.adj) reported in the Results correspond to ANCOM-BC outputs. This description has been added to the Methods.

Figures:

Overall nice figures! I have a few comments/suggestions described below.

Figure 2B: *The text along the top is cut off (e.g. "All_Ag2_breastmilk_complement" reads "g2_breastmilk_complme").*

Figure 2D: *Text legend and the text in the results section say there is one significant difference, but there's no asterisk or indicator to show which one. I can guess based on the figure, but you could add an asterisk to highlight.*

Figures 2B and 2D have been adjusted as suggested: the truncated text along the top of Figure 2B has been corrected, and an asterisk has been added in Figure 2D to indicate the significant difference described in the Results and legend.

Figure 3A: *The figure legend says this is stratified by kennel, litter, and life phase in the text, yet the barplot appears to only be divided by life phase/age.*

The legend of Figure 3A has been corrected to accurately state that the bar plots are stratified by life phase/age, rather than by kennel and litter.

Figure 3B: *Is this showing taxa from across all samples (all kennels, litters, ages, etc.)? I would clarify in the legend. Also, a minor note, but I cannot tell if the smaller dots in the lower right are black (representing a different genus), or if these are blue and still represent Blautia, just*

appearing darker because of the density. Lastly, for genera in both A and B, it would be great if they could have the same colours in both figures to avoid confusion. For instance, *P. hiranonis* is a blue/purple colour in panel B, but *Peptacetobacter* is green in A. If possible to make these consistent, it would improve clarity. Not a major comment if this is not possible.

Figure 3B indeed summarizes taxa across all samples (all kennels, litters, and ages), and we have clarified this explicitly in the figure legend. We have also harmonised the colour scheme between panels A and B so that genera share the same colours across both panels (e.g. *Peptacetobacter hiranonis* / *Peptacetobacter*). In addition, we corrected a minor export-related issue that affected the alignment of the colour legend in Figure 3A. The revised figure now displays consistent and clearly distinguishable colours in all panels.

Figure 3C: *If space, one suggestion is to add a key for the kennel colours as well. Again, this is certainly not a major issue, so okay if not feasible. I can imagine a couple scenarios where that might be helpful for the reader.*

In Figure 3C, the outer rim encodes the test statistic (whether a given taxon is enriched in any kennel), rather than specific kennel identities; therefore, a kennel colour key does not apply in this case. To improve clarity, we have refined the legend to better explain the meaning of the outer rim.

Figure 6: *Is it possible to include an NMDS showing the individual mothers in different colours, with different shapes (or whatever you like) indicating the reproductive state (e.g., normal, first trimester, lactation, etc). This would help clarify how many different mothers were sampled, which is currently unclear. Also, it would show how much variation there is within vs between individuals, which is great information. For instance, even as an individual mother shows these shifts throughout reproductive states, does her microbiome stay more similar to herself throughout that process, or in contrast, does she cluster more closely with another mother in the same reproductive state. My guess is the former, but a simple figure could provide great insight here.*

We fully agree that visualising individual mothers with both individual-level colouring and reproductive-state shapes provides important context for interpreting within- versus between-

mother variation. We have therefore added a new panel (Fig. 6C) following your recommendation.

In this NMDS plot, each mother is represented by a distinct color, and reproductive states (normal, pregnancy, postpartum day, lactation) are indicated by different shapes. As expected, samples cluster more strongly by mother identity than by reproductive state, indicating that inter-individual differences outweigh the effects of reproductive phase. Nevertheless, most mothers show distinct shifts between pregnancy/pre-/postpartum and lactation/normal states, although these shifts do not follow a uniform direction and are not associated with kennel membership.

This figure highlights exactly the biological point you mentioned—mothers tend to remain more similar to themselves across reproductive stages than to other mothers in the same stage. We agree that a larger, dedicated cohort would allow deeper exploration of these dynamics, and we mention this in the Discussion.

Self-Correction

In line 639 of the originally submitted manuscript, in the legend of Table 1, we made an unintentional typographical error in the table number, written as “1B”. In this revised version, we have corrected it to “Table 2”.

Re: mSystems01279-25R1 (Longitudinal Long-Read Microbiome Profiling in a Canine Model Reveals How Age, Diet, and Birth Mode Shape Gut Community Dynamics)

Dear Dr. Dóra Tombácz:

Your manuscript has been accepted, and I am forwarding it to the ASM production staff for publication. Your paper will first be checked to make sure all elements meet the technical requirements. ASM staff will contact you if anything needs to be revised before copyediting and production can begin. Otherwise, you will be notified when your proofs are ready to be viewed.

Sincerely,
Tricia Van Laar
Editor
mSystems

Reviewer #2 (Comments for the Author):

I sincerely thank you for your thorough and thoughtful responses to my suggestions. This is a lovely study and a beautiful paper!